# Drought and Human Impacts on Land Use and Land Cover Change in a Vietnamese Coastal Area

**Hoa Thi Tran** [1,2,†,‡,*] , **James B. Campbell** [1,‡] , **Randolph H. Wynne** [1,‡] , **Yang Shao** [1,‡] and **Son Viet Phan** [2,‡]

1   Virginia Polytechnic Institute and State University (Virginia Tech), Blacksburg, VA 24061, USA; jayhawk@vt.edu (J.B.C.); wynne@vt.edu (R.H.W.); yshao@vt.edu (Y.S.)
2   Hanoi University of Mining and Geology, Hanoi, Vietnam; phanvietson_humg@yahoo.com.vn
*   Correspondence: hoatran@vt.edu or tranthihoa@humg.edu.vn; Tel.: +1-540-449-8146
†   Current address: Geography Department, Virginia Tech, Blacksburg, VA 24061, USA.
‡   These authors contributed equally to this work.

**Abstract:** Drought is a dry-weather event characterized by a deficit of water resources in a period of year due to less rainfall than normal or overexploitation. This insidious hazard tends to occur frequently and more intensively in sub-humid regions resulting in changes in the landscape, transitions in agricultural practices and other environmental-social issues. The study area is in the sub-humid region of the northern coastal zone of Binh Thuan province, Vietnam—Tuy Phong district. This area is indicated as a subject of prolonged droughts during 6-month dry seasons, which have occurred more frequently in recent years. Associated with economic transitions in agricultural practicing, urbanization, and industrialization, prolonged droughts have resulted in rapid changes in land use and land cover (LULC) in Tuy Phong, especially in three coastal communes: Binh Thanh, Lien Huong, and Phuoc The. A bi-temporal analysis using high-resolution data, the 2011 WorldView2 and the 2016 GeoEye1, was examined to assess LULC changes from observed severe droughts in those three communes. Results showed a dramatic reduction in the extent of hydrological systems (about 20%), and significant increases of tree canopies in urban areas and near the coastal areas (approximately 76.8%). Paddy fields declined by 51% in 2016; such areas transitioned to inactive status or were alternated for growing drought-tolerant plants. Shrimp farming experienced a recognizable decrease by approximately 44%. The 2014 map and field observations during summer 2016 provide references for object-based classification and validation. Overall agreement of results is about 85%.

**Keywords:** Vietnam; Binh Thuan; sub-humid region; drought-human impacts; LULC; high-resolution images; object-based classification

## 1. Introduction

Drought is defined as an extended period of precipitation deficiency, which leads to severely arid conditions during dry seasons [1]. Drought has considerable impacts on ecosystems, environment, and society. For example, drought may cause changes in landscapes as a formation of dry area, and grassland, and a promotion of aeolian processes. Otherwise, drought also leads to a reduction of water quality and quantity [2,3], soil moisture, and soil productivity. Reduction of soil moisture cannot guarantee soil microbial activities, which play an essential role in organic matter decomposition and synthesis to create soil fertility. Therefore, soil is very sensitive, and easily desiccated. Fresh water is very important not only for domestic uses, farming, and grazing, but also for preserving natural habitats. Sudden decreases of water in hydrological systems due to severely arid conditions can affect both biotic and abiotic components of the ecosystem [4]. Plants and vegetation are vital components

that create biomass for the ecosystem, but under drought conditions, loss of soil moisture, and infertile soils, survival is not secured. There will be a risk of bare land and aeolian landscapes, which, coupled with unsustainable land management, promote desertification, complicating land productivity recovery. Drought also has substantial impacts upon societal and economic priorities, including increased budgeting for fresh water, reduction of crop yields, losses from businesses, families, and government [2]. Migration from dry areas is often forced by drought, which can drive land use transitions and changes in LULC [5]. However, water overuse, overpopulation, and mitigation have certain impacts of deficits of both surface and ground water resources resulting in more severe hydrological drought [3,6].

Dry areas in the tropics are characterized by prolonged dry seasons under conditions of low humidity, high air temperatures, and drying up of seasonal rivers and streams. Instead of precipitation deficits, known as a cause of meteorological and agricultural drought, there is here a linkage between drought and human activities [7]. Humans, via their activities, have created negative influences on water cycles, such as deforestation, urbanization, and intensified/extensified farming [8]. Such activities increase transpiration, evaporation, and reduce water-holding capacities of soil. Otherwise, dam construction to regulate hydrologic systems, to mitigate floods, to produce hydro-electric power, and to irrigate farmlands is, on one hand, necessary with respect to sustainable land management; on the other hand, it has unfortunate impacts on downstream surface water flow, and creates higher demands upon water supply [3]. Generally, a long drought duration, and its interactions with human activities, has directly or indirectly responded to changing landscape and changes of LULC. Understanding impacts of drought-human interaction is very important to govern water restrictions and preserve soil quality in the condition of drier-than-normal conditions. Furthermore, clarifying drought impacts on LULC changes can help to support sustainable land management in dry lands.

In this study, we examine changes in LULC upon drought events and human activities in three coastal communes of Tuy Phong district, Binh Thuan, Vietnam. The study area is shown in Figures 1 and 2 covering three communities: Phuoc The (in the north), Lien Huong (in the middle), and Binh Thanh (in the south). We were seeking answers of our three research questions regarding LULC changes during drought periods: (1) How has LULC changed during drought events of our study's five-year inquiry, 2011–2016? (2) How did drought events affect specific land uses, such as agriculture, or shrimp farming? (3) How did local inhabitants manage land use during those drought events? Thus, to reveal both impacts of drought and humans on LULC in general, and some specific types of land use in particular, we proposed applying high-resolution images in LULC change assessment at two levels of detail. The first level is land cover, which observes any change in seven main types: built-up land, open water, agriculture, salterns and shrimp farming, vegetation, bare land, and sand. The second level is more detailed, consisting of 13 classes, and focuses on specific land use types that reveal human impacts by adding five more classes: (a) active and (b) inactive agriculture (paddy fields, orchards and inactive fields); (c) active and (d) inactive shrimp farming; and (e) vegetation on sand. Descriptions of imagery data and land types are represented in Tables 1 and 2 in Section 3.2; imagery datasets are shown in Figure 2, and training samples for each land type are in Appendix A.

## 2. Study Area

### 2.1. Study Area Background

Tuy Phong is a Northeastern district of Binh Thuan province, south-central coast of Vietnam. Although Tuy Phong is positioned in the tropical wet–dry climate following the Köppen-Geiger classification characterized [9], annual precipitation is quite low compared to other parts of Binh Thuan. Northern districts, such as Tuy Phong and Bac Binh, are much drier than other regions; Southern districts such as Ham Tan and Ham Thuan Nam are wetter [10]. Figure 3 shows annual rainfall in our study area in comparison to other part of Vietnam which is considered to be one of two

driest regions in the whole country. Annual precipitation of the Northern area is less than 800 mm per year; in some years, it can be less than 250 mm; the driest period is from January to March (monthly rainfall less than 4 mm—see Figure 3, top right chart, data were collected at Lien Huong weather station). There is a distinction of rainfall distribution between wet and dry seasons in Tuy Phong: rainfall mostly occurs during wet season (from May to October), while there is less than 50 mm or no rain during the 6 months of dry season (from November to April). Thus, preserving water storage and practicing irrigation in this district during dry season needs to be taken seriously. However, in the Southern part, annual precipitation is on average 1700 mm per year, and can exceed 2500 mm per year, which is one beneficial condition for high water demand crops such as rice.

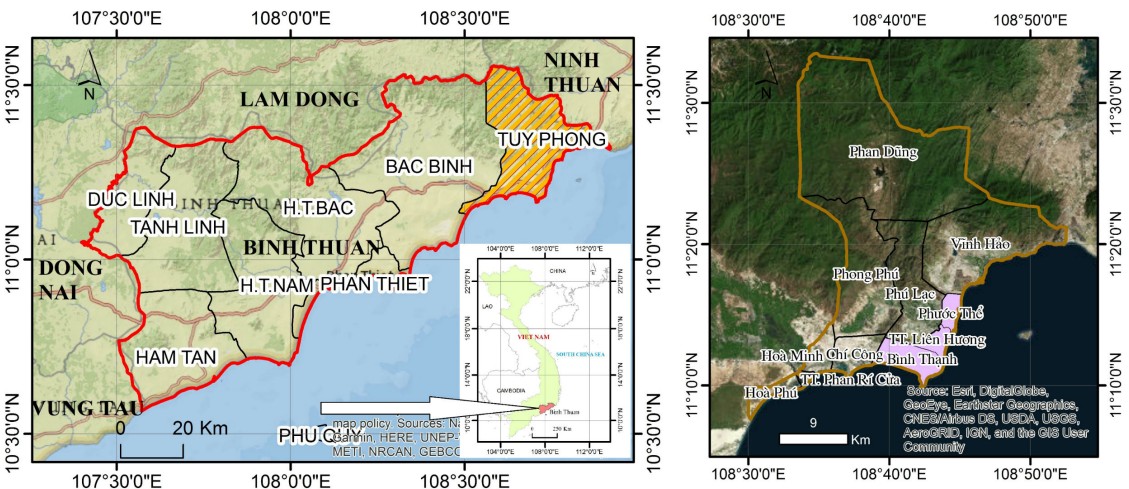

**Figure 1.** The case study is covering three coastal communities Binh Thanh, Lien Huong, and Phuoc The of Tuy Phong District which are highlighted in pink on the right map.

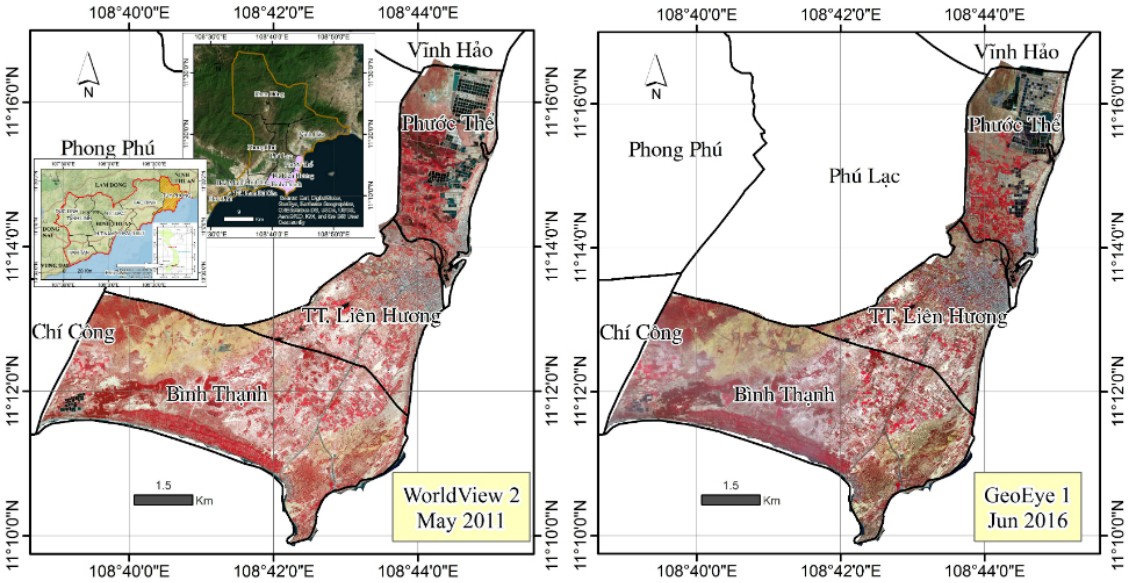

**Figure 2.** False color band combination for two-image datasets used for the research area: the 2011 WorldView2 (NIR2, R, G as R,G,B), and the 2016 GeoEye1 (NIR, R, G as R,G,B). See Table 1 for the sensor-specific band number designations along with bandwidths

Population of this district has been increasing rapidly, doubling in the 13 years from 90,000 in 2002 to 188,000 in 2015. High population creates pressure for local food production, demand for fresh

water, health care, and education, but on the other hand, it can provide an abundant labor force for the local labor market. However, most of the local labor force in Tuy Phong is involved in farming and in salt production, which produce less economic value [11], and brings lower family income. Furthermore, farming and salt production are water and climatic dependent. As a result, droughts and water shortages may directly threaten local people.

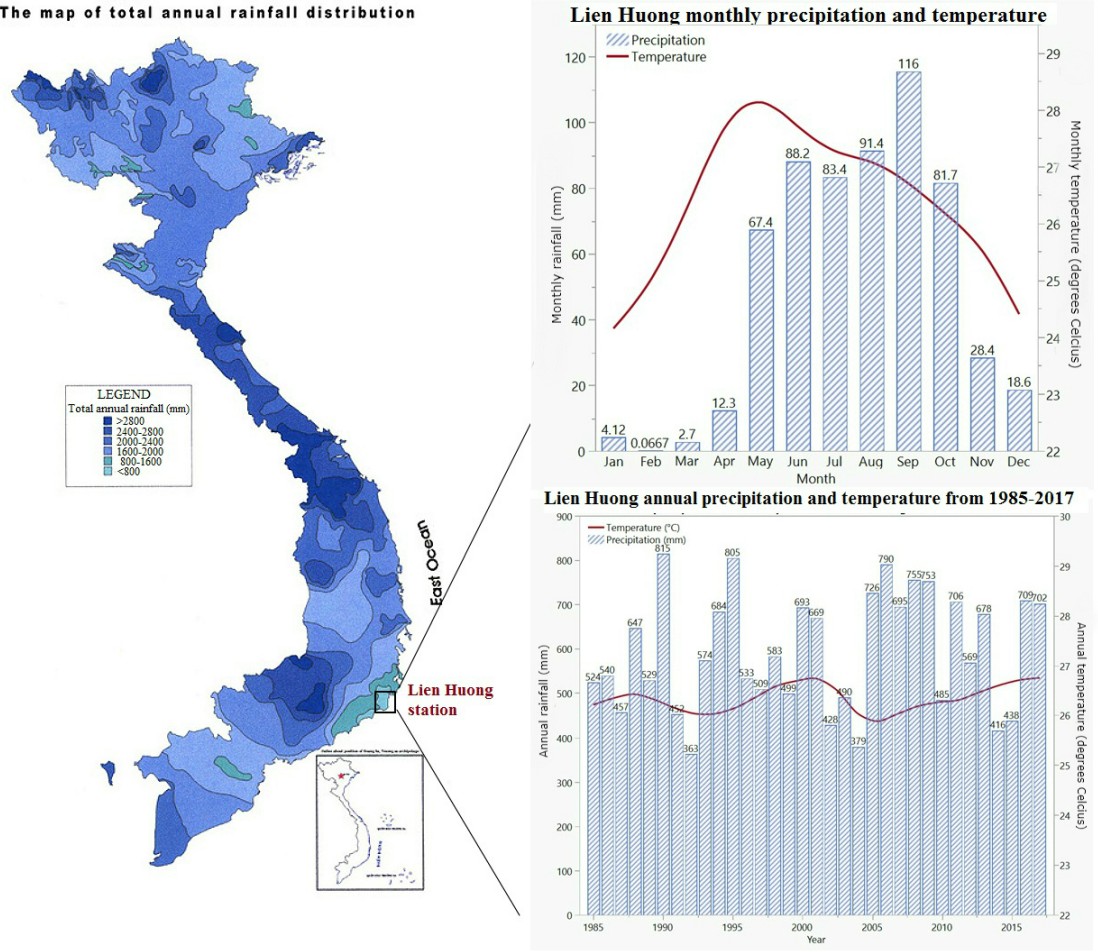

**Figure 3.** Map of total annual rainfall distribution of Vietnam (source: *The center of database establishment for Environmental resources in Vietnam, Geography institute (1999)* [12]; and charts of monthly and annual precipitation and temperature report at Lien Huong station during 1985–2017 (meteoblue.com [13]).

Water shortage is one of major issues in Tuy Phong, due to the occurrence of prolonged meteorological drought regarding rainfall deficits during dry seasons, and overuse of local residents [14,15]. The lower right chart in Figure 3 illustrates fluctuations of annual precipitation and annual temperature acquired at Lien Huong station during 1985–2017; this station is in Lien Huong commune, one of three coastal communes involved in this study. Average precipitation of this 32-year observation is 612 mm per year. During 1990–2005, most of yearly rainfall is less than this value, most significantly in 1992 (363 mm), 2002 (428 mm), and 2004 (379 mm), respectively. Otherwise, in recent years, such as 2010, 2012, 2014, and 2015, annual precipitation also denoted lower values than the average one. These meteorological data were collected at meteoblue.com. Upon local reports, during the 2014–2015 dry season, there was a significant decline of water in the main reservoirs; in 2015, Da Bac Lake lost 90% of its fresh water [16]. Our recent study of agricultural drought severity in Tuy Phong district under a scenario of rainfall shortage indicated that those coastal communes are sensitive subjects to drought at moderate to mild levels [17], which are shown in light green to yellow

in Figure 4. The analysis of that study was based on an idea of using the Vegetation Health Index (VHI) extracted from satellite images [18], which are Landsat time series data to access spatial and temporal agricultural drought patterns in Tuy Phong.

Another consequence of frequent drought events is LULC changes. Due to water shortages, irrigated and rain-fed lands for paddy fields or rice production in Tuy Phong have transitioned to inactive status. Additionally, high temperatures associated with drought may disturb shrimp farming. Figure 2 shows a significant reduction of agricultural lands in Phuoc The commune; a majority of agricultural lands occurred in a reddish color on the WorldView2 image was replaced by dry lands (brownish color on the GeoEye1). Furthermore, there was an expansion of dry salterns and dry shrimp fields in this northern commune's coastline on the 2016 GeoEye1 image shown in tan color. Drought also promotes advances of mobile sands, and creation of land degradation processes, through removal of top soil, and decreasing soil fertility. Advances of sand dunes also influences local people's normal life as it buries roads, buildings, and vegetation; the right image in Figure 5 shows that one part of Lien Huong communal roads was dominated by red and yellow sand. On images (Figure 2), mobile sand shown in white color dominants areas between Binh Thanh and Lien Huong communes, and coastal areas.

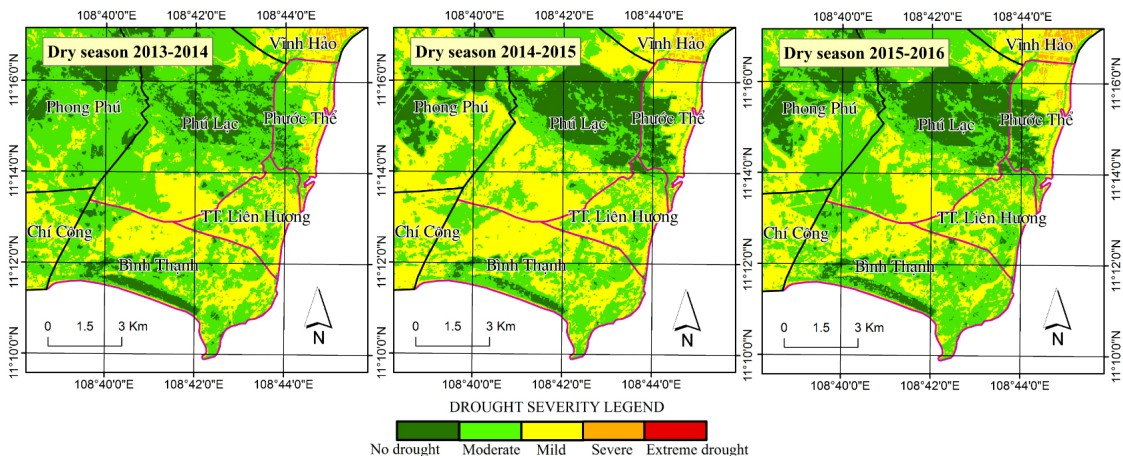

**Figure 4.** Agricultural drought severity during dry seasons from 2013 to 2016 in the study area (red color border) in Tuy Phong district, Binh Thuan. The study area was indicated under moderate to mild impact of drought. The analysis was based on assessing spatial and temporal Vegetation Health Index distribution via Landsat images over 27 years [17].

In addition to drought impacts, local residents in Tuy Phong have significant influences on LULC changes via land-economic transition, and drought-adaptive mitigated practices. Many policies have been designed to protect land, and prevent salinization, such as constructing man-made dams and lakes for water storage and irrigation improvement, or cultivating unused land, or separating agriculture from salt-shrimp production areas. Da Bac Lake was built in 1990 just about 2 km away from salt production; it has played an important role in storing and governing fresh water for irrigation in Vinh Hao commune. The Long Song Dam, one of the biggest man-made dams in Viet Nam, was constructed in 2000 in Phong Phu commune [19] to irrigate more than 4000 ha of cultivated land, to supply fresh water for households, and mitigate food in lower areas. Other cultivated land near the coast is mainly rain-fed. Urbanization and industrialization mainly occur in Tuy Phong's coastal area. New power plant construction includes: the Vinh Tan thermal power plants in 2010 in Vinh Hao commune, and Binh Thanh's wind power plants to use an abundant ocean wind resources in 2008. Tourism, factories, and marine fishing have encouraged local residents to migrate to the coast.

## 2.2. Agricultural Practicing During Drought Period

Agricultural areas of Tuy Phong are mainly located near water resources and the coastline as flat terrain. However, due to dramatic weather (long dry seasons, lack of rainfall, high temperatures, and high wind speeds), infertile soil, insufficient irrigation, and aeolian erosion, local agriculture has faced many difficulties. Many of the surface soils of coastal areas of Tuy Phong and Binh Thuan province contain minerals and heavy metals, such as iron, which are visible as yellow and red surface soils. Ocean winds also displace topsoil and transport sand from coastlines and dunes to inland sites. As a result, such soils are now largely infertile, unable to support agriculture. Furthermore, such soils are very sensitive to land degradation and desertification. For example, the left image in Figure 5, acquired in summer 2016, shows a cassava field—one type of drought-tolerant plant in Binh Than commune grown on orange soil representing the high amount of iron contamination. Near the coast, there are wind farms which have been established since 2011. Wind power fields were used to practice agriculture in the past; currently they are bare and unused. On the research images, those soil types are located mainly in the Northwest and Southeast of Binh Thanh shown in a yellowish color (Figure 2).

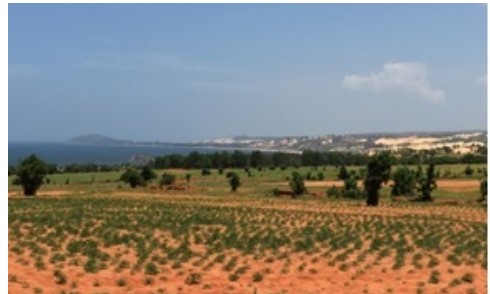 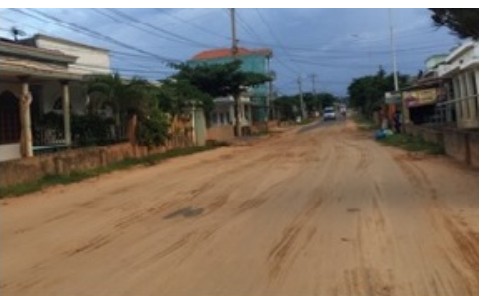

**Figure 5.** Cassava field in Binh Thanh commune (**left**); and sand-covered road near the coast in Lien Huong commune (**right**).

Dramatic reductions of supplies during prolonged droughts, infertile soils, and land degradation have changed status of both irrigated and rain-fed fields from active to inactive, both temporarily and permanently. Local residents here have applied crop rotation, planted cover crops, and installed new irrigation systems to mitigate increasing aridity. There are replacements of paddy fields by orchards, salterns, and shrimp fields. Dragon fruit (pitaya), grape, and cassava replace the rice base on their drought-tolerant characteristics, and their higher economic value. However, many fields have maintained their inactive status for many years, especially rain-fed fields in Binh Thuan and Lien Huong commune. Transitioning agricultural land to salterns or shrimp fields on one hand may bring higher income to farmers, but on the other hand, it can affect soil fertility (salinity intrusion), promote soil erosion, and indirectly enhance sand-dominant and land degradation processes in neighborhood areas. Furthermore, under long-term aridity, without sustainable practices or cultivation, inactive fields near the coast are directly subject to land degradation and desertification.

## 3. Materials and Methodologies

### 3.1. Datasets

To carry out this research, we acquired WorldView2 and GeoEye1 satellite images—both providing high-resolution multi-spectral imagery of our region which are shown in Figure 2. The WorldView2 image was captured on 22 May 2011 providing eight spectral bands, and one panchromatic band, at 1.84 m and 0.46 m spatial resolutions, respectively. The GeoEye1 image, on the other hand, was recorded on 16 June 2016 with four spectral channels and one panchromatic band at 1.65 m and 0.41 m spatial resolution. Table 1 shows the numbers and range of wavelengths of each spectral channel of

two images using in this study. Prior to classification, images were calibrated and enhanced. Digital Globe Courtesy kindly contributed their image datasets for this research.

**Table 1.** WorldView2 and GeoEye1 bands.

| No | Band | WorldView2 | Geoeye1 |
|----|------|------------|---------|
| 1 | Coastal | 400–450 nm | NA |
| 2 | Blue | 450–510 nm | 450–510 nm |
| 3 | Green | 510–580 nm | 510–580 nm |
| 4 | Yellow | 585–625 nm | NA |
| 5 | Red | 630–690 nm | 655–690 nm |
| 6 | Red Edge | 705–745 nm | NA |
| 7 | NIR1 | 770–895 nm | 780–920 nm |
| 8 | NIR2 | 860–1040 nm | NA |

*3.2. Image Analysis Procedures*

The procedure of image processing and analysis is represented shortly in the flowchart—Figure 6. Generally, the preprocessing procedure carried out processes of image calibration, radiometric—spectral enhancement, pan-sharpening, and sub-setting, then images were classified to match land use classes, and, finally LULC maps were generated and validated. The 2014 map made by the government, Google Earth images, and several land types' samples of 2016 field research are used for selecting the training samples, and decision making for the manual editing (shown in Figure 6). To validate our results, we used Google Earth images and ground reference points collected during our field study in 2016.

To classify the image data, we applied object-based image classification method, which is commonly applied for high-resolution images [20,21]. This method is based on distinctive characteristics of each object, such as color, shape, size, and relationships to others to define LULC [22–24]. As mentioned in the Introduction, there are two categorized systems used to classify LULC types at two detailed levels, Table 2 describes those two systems and identities of land types. In this research, training datasets were sampled mainly based on each object's characteristics. Besides, we investigated some band indices that supported decision rules of assigning classes. These indices are tremendously supportive for making decisions on feature assignment procedure [25–28].

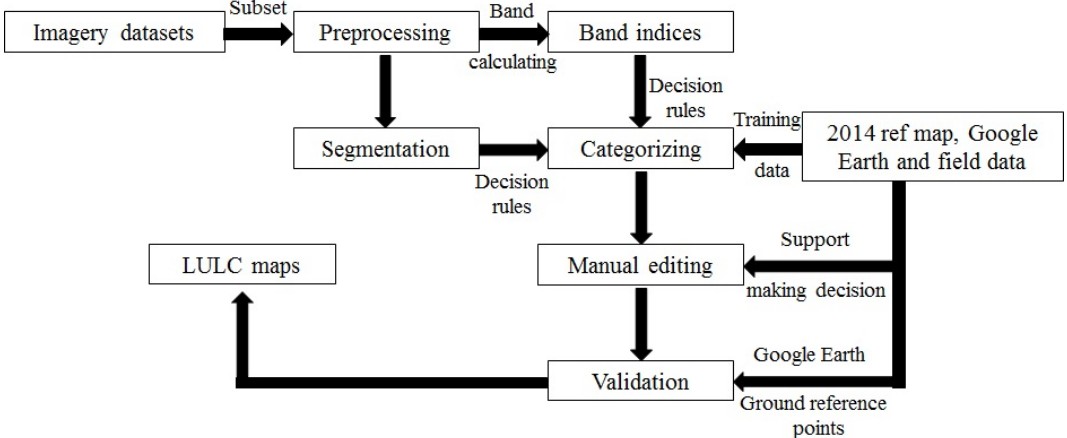

**Figure 6.** Flowchart of image analysis and maps generalization procedure.

Four band indices—Normalized Difference Vegetation Index (*NDVI*), Normalized Difference Water Index (*NDWI*), Normalized Difference Soil Index (*NDSI*), and *NHFD* (Non-Homogenous Feature Difference)—were involved in the procedure of classifying the WorldView2 dataset using its 8

spectral channels in advance. The *NDVI* was approached to classify the GeoEye1 as its four available spectral bands (shown in Table 1). Formulas * to calculate those indices are presented below. *NIR*1 and *NIR*2 are two Near Infrared channels, their wavelengths are shown in Table 1.

$$NDVI = \frac{Red - NIR2}{Red + NIR2} \quad \text{or} \quad NDVI = \frac{Red - NIR}{Red + NIR} \tag{1}$$

$$NDWI = \frac{Coastal - NIR2}{Coastal + NIR2} \tag{2}$$

$$NDSI = \frac{Green - Yellow}{Green + Yellow} \tag{3}$$

$$NHFD = \frac{RedEdge - Coastal}{RedEdge + Coastal} \tag{4}$$

*\* These formulas are suggested for a World View 2 object-based classification by Antonio Wolf, Digital Globe Foundation [29].*

Training samples, suggested band indices and typical characteristics of each class are described in the Appendix A—Table A1.

**Table 2.** Description of land use and land cover categorized system in this research at two detailed levels.

| ID | Level 1 Land Type | ID | Level 2 Land Type | Description |
|---|---|---|---|---|
| 1 | Built up | 1 | Built up | Urban area, transportation, mining and windpower. |
| 2 | Salterns and shrimp farming | 2 | Salterns | Used for sea salt production; near coastline; different shape |
| | | 3 | Active shrimp farming | Square objects; used for shrimp cultivation, well organized. |
| | | 4 | Inactive shrimp farming | Square shapes; no practice at time of observation. |
| | | 10 | Land in shrimp farming | Normally linear object;bright color (tan); used for transportation. |
| 3 | Water | 5 | water | Lakes, ponds, streams, rivers. |
| 4 | Agriculture | 6 | Rice production /paddy fields | In different shape (rectangular); smooth surface; still practicing. |
| | | 11 | Orchards | Drought-tolerant plants; normally in rectangular; clear rows, and pots; mixing urban (single buildings or roads). |
| | | 12 | Inactive agriculture | No/sparse vegetation; normally rectangular; clear/sharp borders. |
| 5 | Vegetation | 7 | Vegetation | Near urban area, or fields; dense; rough surface. |
| 6 | Bare land | 8 | Bare land | Bare soil, no/sparse vegetation (small and low bush). |
| 7 | Sand | 9 | Sand | No vegetation;yellow, or white; near the coastline. |
| | | 13 | Vegetation on sand | Small and low bush; near. coastline; dense or sparse. |

## 3.3. Recommendations on Object-Based Classification Procedure

While executing the image classification procedure applying the object-oriented method, there were several emerging issues resulting in a lot of time consumption. For example, reselecting train samples was attempted several times to replace unsatisfactory old ones. Finding optimistic numbers

of pixels for the segmentation procedure also required trials on differentiated objects. Some objects share similar characteristics such as color or shape, so an additional procedure needed approaching. Figure 7 is an example of how the classification procedure was conducted based on the object-oriented method. There are several classes that need to be reclassified due to an issue of "mixed pixel"—pixels belonging to built-up areas were categorized into water, for instance. In this circumstance, to resolve the mixed-pixel problem, a rule that all areas that were less than 1000 pixels should be built up was derived. Instead of using the area of objects, other characteristics can be used, such as color, associated index (*NDVI*), or training samples. Involving each object's characteristics rather than isolatedly using each pixel's value in the classification procedure is one of significant advantages of the object-oriented method compared to the general supervised classification. However, there are some highlighted points that should be considered, which are represented below:

- Shadowed areas: On GeoEye 1 image, we observed shadows of vegetation and low buildings. In this circumstance, this shadow effect was not significant, so no further procedure was applied. However, if shadow effects are obvious (mainly in urban or forested areas), an object—spatial relationship (nearest neighbor, for instance) will be applied in the decision rules.

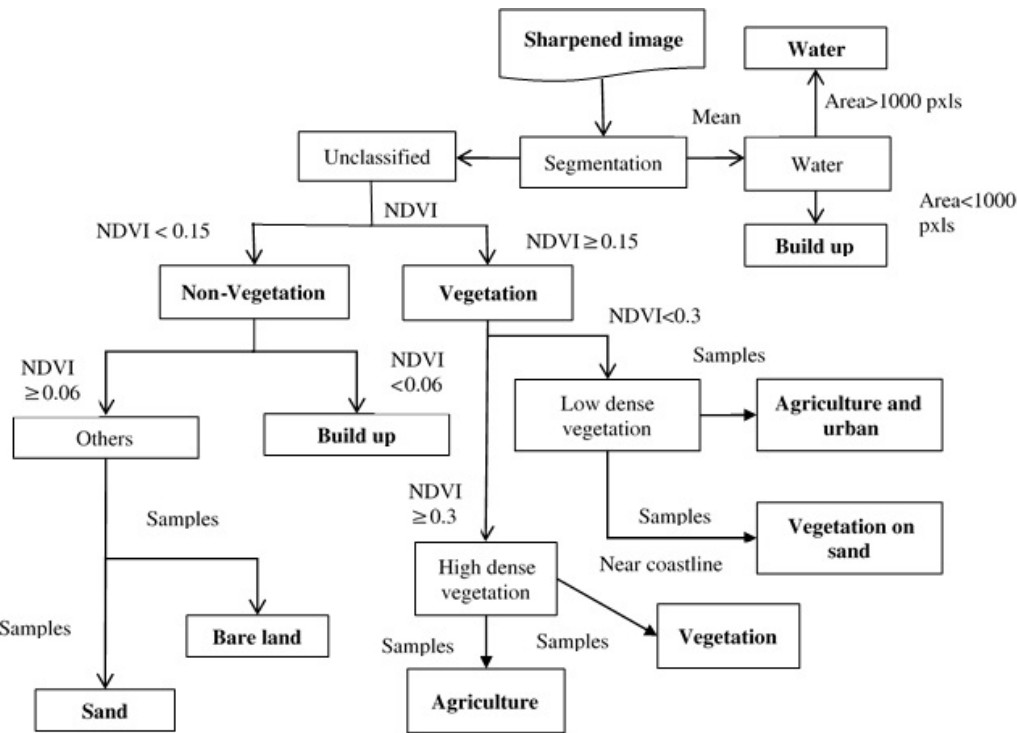

**Figure 7.** An example of classification hierarchy (decision tree) to categorize objects into classes of the Lien Huong GeoEye1 image.

- Band indices: Using different band indices is a good approach to derive effective rules of decision tree to assign classes. It required less time to categorize classes on the Worldview2 image than the GeoEye1 one, especially when it is effective to differentiate roads from water layer. On images, road and water surfaces often appear as similar colors—dark blue. Nevertheless, these indices are quite sensitive, so in each case, the selected range of values of each index designated to each class are different. The image preprocessing step is very important before calculating band indices.
- Segmentation procedure: This procedure requires a lot of time, computer RAM storage, and analyst experience. Sub-divided regions that are too small or too large may increase processing time, or lead to missing data or mixed classes.
- Decision tree: Constructing rules for the decision tree to categorize objects into classes is very important, and obviously not an easy mission. Each object has its own typical characteristics,

and shares some with others. The more indicators there are, the more supportive and successful decision rules are. However, there is another constraint of cost and time consumption.

- Subset images helps to save time to generate classified rules and decision trees. We divided the study area into three parts regarding the administrative boundary.
- Manual editing is a necessary step to approach a better result. A significant difference between object-based classification and supervised classification is the smallest object. A pixel is the smallest object in a supervised method, so "salt and pepper" errors exist on the classified image. An object that may cover at least 50 or 100 pixels, depending on segmentation methods is the smallest one in the object-based classification. Thus, it is easier to detect mis-classified objects on results, and bring them to their true class. Ecognition Developer software allows users to process segmentation, and manual editing very conveniently.

## 4. Results and Discussion

### 4.1. Land Use—Land Cover Classification

To execute this research, the 2011 and 2016 LULC maps were generated at two levels: (1) Level 1 within seven classes assessing general trend of LULC changed during 2011–2016, and cross validating with the 2014 map; and (2) Level 2 with 13 classes assessing drought-human impacts on agricultural and vegetating lands. Our maps, statistical data, and validation results are shown in Figures 8–10, Tables 3–5.

To validate our final results, we conducted contingency tables for LULC maps 2011 and 2016 (summary of the classification accuracy at level 2 is shown at Table 3) by using 521 random points including 460 automatically generated points and 61 ground reference points collected in the 2016 field research of land sampling. Furthermore, Google Earth images was used to cross-validate our images and final maps. Overall accuracies of two-image classification at level 2 are 85% (WorldView2), and 87.3% (GeoEye1). According to the contingency tables, class 7 and class 13, which are urban vegetation and sandy vegetation, respectively, are less accurate in comparison to others. The reasons are: firstly, the segmentation method resulted in a different size of polygon that could not perfectly isolate objects; secondly, some vegetated areas were mixed with agricultural lands, or buildings, or vegetation—sand mixing, so it is not easy to differentiate them correctly. Accuracy of water layers is very low because there are several random points belonging to the water layer (see Table 3). At level 1 classification, the overall accuracies of those two classified images are much higher than 90% as there are fewer categories, and less ambiguity between classes. Thus, in the classification procedure, designed numbers of classes, and class assignment methodologies are the main factors that affect overall accuracy.

### 4.2. Overall Changes in LULC during 2011–2016

At level 1 of image classification, we occupied an overall trend of LULC changes in the case study within seven classes: built up, salterns and aquaculture (shrimp farming), surface water, agriculture, bare land, and sand. Figure 8 and the data Table 4 below show our findings of LULC changes during the interval 2011—2016. Generally, there was a dramatic increase of vegetation cover, especially in urban areas from 475.1 ha (2011) to 839.8 ha (2016), an increase of about 76.8%. Sand and water declined 20% after 5 years, while other land types such as bare land, agriculture, and shrimp farming rose slightly, about 3%. Built-up land decreased by approximately 3% because we only counted impervious surface (buildings, single blocks, or roads). Several types of vegetation cover such as urban vegetation, bushes and plants grown on sandy lands (near the coast) are both categorized into the same class. On the maps shown in the Figure 8, vegetation (dark green) increased both in urban areas (red) and near the coast, replacing sandy lands (light blue). These types of vegetation play important roles to reduce urban heat islands in urban areas, to stabilize sand dunes near the coast, and to protect agricultural lands from sand-dominant processes as a windbreak solution during drought periods and dry seasons.

**Table 3.** Statistical summary from contingency tables generated to assess accuracy of two land use maps in 2011 and 2016. Unit:%.

| ID | Land Types | 2011 Map's Accuracy | | 2016 Map's Accuracy | | Overall |
|----|-----------|---------|----------|---------|----------|---------|
| | | User's | Producer's | User's | Producer's | |
| 1 | Built up | 86.2 | 92.6 | 98 | 84.5 | 2011's overall = 85% |
| 2 | Salterns | 100 | 100 | 100 | 93.8 | 2011's kappa= 82.9% |
| 3 | Active shrimp farming | 100 | 100 | 100 | 100 | 2016's overall = 87.3% |
| 4 | Inactive shrimp farming | 90 | 100 | 83.3 | 90.9 | 2016's kappa=85.6% |
| 5 | Water | 66.7 | 66.7 | 100 | 66.7 | Total points = 521 |
| 6 | Rice production | 74.5 | 87.2 | 84 | 100 | |
| 7 | Vegetation | 84.2 | 61.5 | 79.1 | 89.5 | |
| 8 | Bare land | 76.9 | 81.1 | 87 | 88.9 | |
| 9 | Sand | 93.3 | 79.5 | 100 | 77.1 | |
| 10 | Land for aquaculture | 100 | 83.3 | 92.3 | 92.3 | |
| 11 | Orchards | 91.7 | 64.7 | 83.3 | 71.4 | |
| 12 | Inactive agriculture | 78.5 | 93.9 | 92.7 | 93.6 | |
| 13 | Vegetation on sand | 83.5 | 94.7 | 71.8 | 87.1 | |

**Table 4.** Changes in different LULC types.

| No | Land Types | Area in 2011 (ha) | Area in 2014 (ha) | Area in 2016 (ha) | Difference 2011–2016 (ha) | % of Difference |
|----|-----------|---------|---------|---------|---------|---------|
| 1 | Built up | 418.8 | 725.3 | 406.5 | ↓ 12.3 | ↓ 2.9 |
| 2 | Salterns and shrimp farming | 311.9 | 321.9 | 322.4 | ↑ 10.5 | ↑ 3.4 |
| 3 | Water | 30.8 | 17.2 | 24.8 | ↓ 6 | ↓ 19.5 |
| 4 | Agriculture | 1229.4 | 2293.3 | 1269.3 | ↑ 39.9 | ↑ 3.2 |
| 5 | Vegetation | 475.1 | 517.9 | 839.8 | ↑ 364.7 | ↑ 76.8 |
| 6 | Bare land | 291.7 | 156.3 | 300.6 | ↑ 8.9 | ↑ 3.1 |
| 7 | Sand | 1992.5 | 755 | 1592.5 | ↓ 400 | ↓ 20.1 |

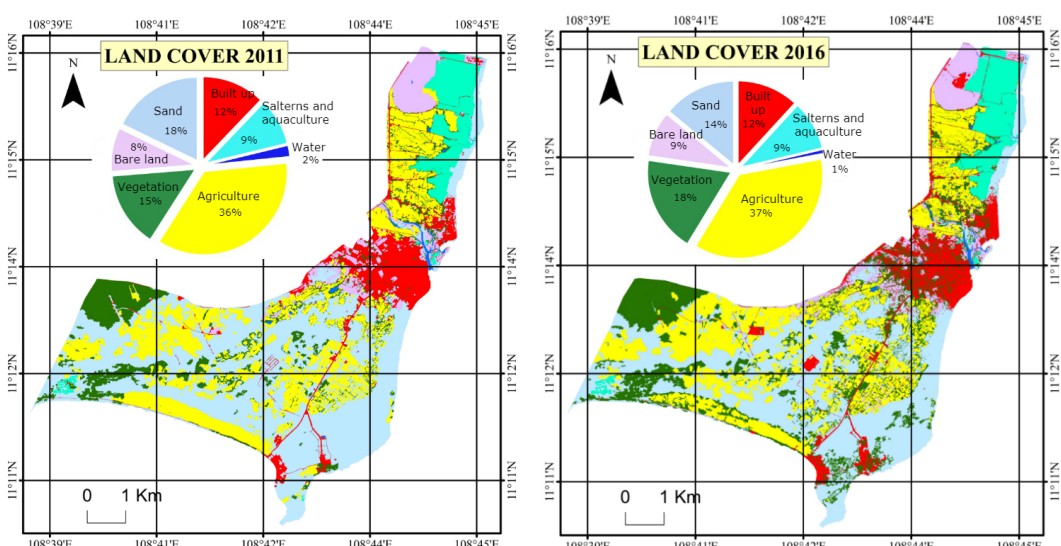

**Figure 8.** Land cover maps pie charts of 2011 and 2016.

*4.3. Impacts of Drought and Human on LULC Change during 2011–2016*

The second level of classification was focusing on revealing the human role on LULC change under prolonged drought occurrences in the study area. We investigated specific land types such as agriculture, salterns, urban vegetation, and sandy vegetation, which were under high impact of local people via land practicing. Figure 10 is an overall report of LULC change by local resident influence.

#### 4.3.1. Agricultural Land Transition

Droughts have had impacts on the environment and society [7,30,31]. They can disturb agricultural activities via fresh water shortages; they may interrupt shrimp farming because higher water surface temperatures simulate evaporating processes. However, high air and surface temperature, lack of rainfall, and more solar radiation, which are climatic characteristics of droughts, stimulate water-vaporing processes, and as a result, marine salt production may get more benefits. Humans, via their land practicing methods, also change use purposes of land to adapt to the change of weather both in the long term and short term depending on the duration of extreme weather (drought) and irrigation conditions. Figure 9 illustrates desiccated paddy fields in Phuoc The due to insufficient irrigation; the image was taken in July 2016 in the middle of wet season. To understand how droughts and humans affect LULC, we accounted for changes in agriculture and shrimp farming by defining more precise land use classes: rice production, inactive agriculture, orchards, salterns, active shrimp farming, and inactive shrimp farming (level 2 in Table 2). A further analysis applied to Phuoc The commune in Section 4.4 shows a "closer look" on these influences.

From Table 5, from 2011–2016, the area devoted to rice production declined by more than 50%. Inactive agriculture (increased 32%), bare land, and orchards replaced this 50% decrease. The area devoted to orchards was 76.9% greater in 2016 than in 2011. The transition from rice to dragon fruit, grape, and cassava is one of the examples of applying adaptive methods under drought impacts of local residents. Those types of plants require less water consumption than rice. Moreover, dragon fruit and grapes are more valuable than rice, and can reproduce over many years. Inactive agricultural area was divided into two types: (1) temporal interruption due to water shortage (mainly found in Phuoc The); and (2) permanent interruption (other communes).

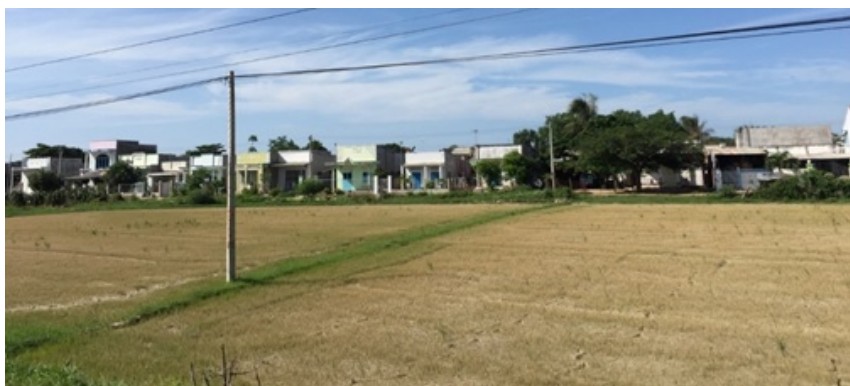

**Figure 9.** Lack of fresh water for irrigation created these desiccated lands in the commune of Phuoc The.

Long-term inactive agriculture is one of risks of land degradation, which is found mostly in Binh Thanh and Lien Huong communes where fields are mainly rain-fed. Because of the lack of precipitation and insufficient irrigation, these lands cannot support agriculture, based on our field observation, and using Google Earth time series, they have not been cultivated for more than 20 years. Those inactive agricultural lands are typically orange in appearance on standard false color composites, which reveal the occurrence of iron in the soil. Characteristics of soil and water shortages due to frequent droughts are two main factors that lead to land degradation or desertification in the study area.

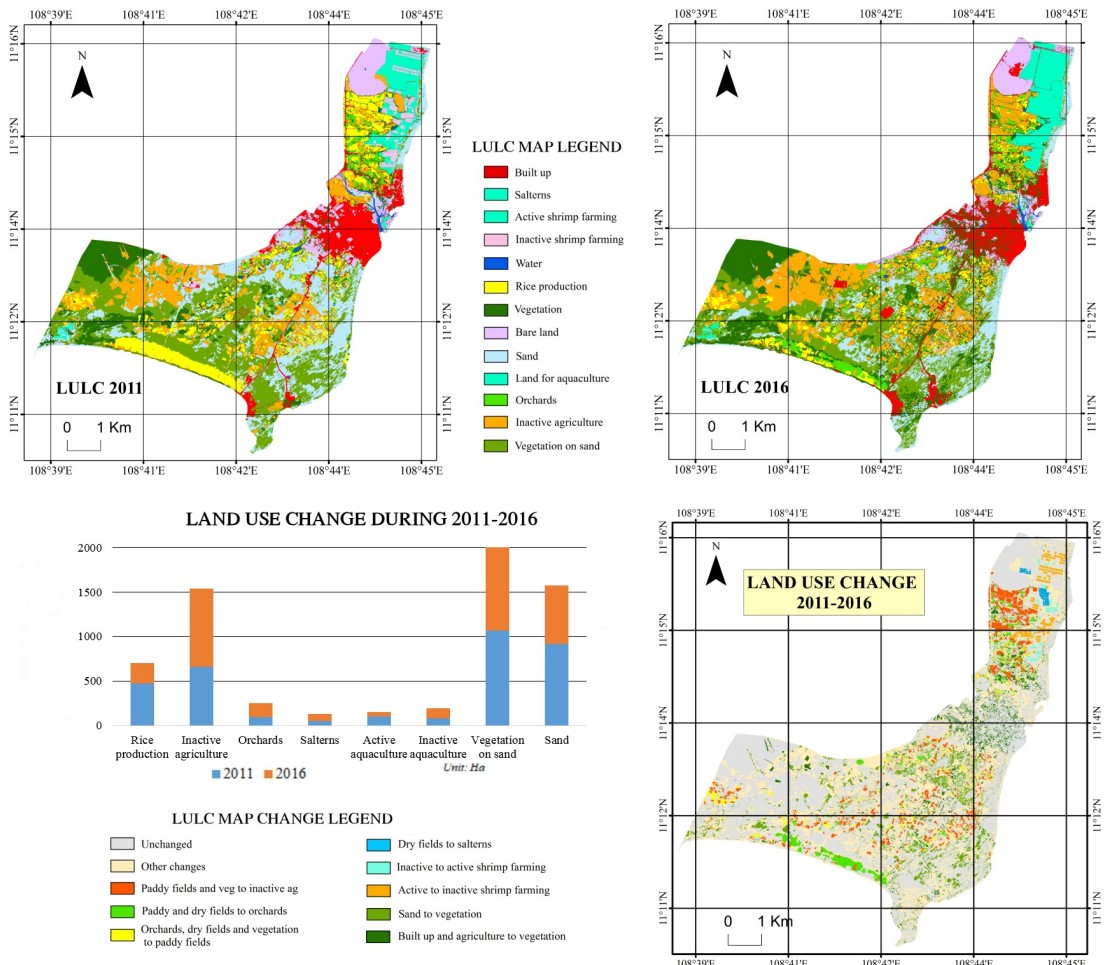

**Figure 10.** Maps of land use and land cover change during 2011–2016 .

**Table 5.** Changes of specific land types regarding to agriculture and shrimp farming during 2011–2016.

| No | Land Types | Area in 2011 (ha) | Area in 2016 (ha) | Area of Difference (ha) | % of Difference |
|---|---|---|---|---|---|
| 1 | Rice production | 474.6 | 228.5 | ↓ 246.1 | ↓ 51.9 |
| 2 | Inactive agriculture | 662.2 | 877 | ↑ 214.8 | ↑ 32.4 |
| 3 | Orchards | 92.6 | 163.8 | ↑ 71.2 | ↑ 76.9 |
| 4 | Salterns | 50.4 | 80.4 | ↑ 30 | ↑ 59.5 |
| 5 | Active shrimp farming | 99.8 | 55.8 | ↓ 44 | ↓ 44.1 |
| 6 | Inactive shrimp farming | 79.5 | 114.7 | ↑ 35.2 | ↑ 44.3 |
| 7 | Vegetation on sand | 1071.9 | 937.7 | ↓ 134.2 | ↓ 12.5 |
| 8 | Sand | 920.6 | 654.8 | ↓ 265.8 | ↓ 28.9 |

### 4.3.2. Salterns and Shrimp Farming

Shrimp is more valuable, but it requires a big investment in prawn shrimp, food, and cleaning, etc. Moreover, shrimp is sensitive to environmental conditions, such as water temperature, water quality, and diseases. Shrimp farming in Tuy Phong is mainly based upon individual operators, so farmers have faced many difficulties, especially finding markets for their products, and reinvestment after losses. In the study area, salterns and shrimp farming are located in Phuoc The commune (cyan color on LULC maps—Figure 10). During the period 2011–2016, there was a slight increase in salterns and shrimp farming. From Table 5, it can be seen that active shrimp fields declined by nearly 40% in 2016 compared to 2011; they were replaced by dry fields. A local report described this disruption: regarding the Tuy Phong agricultural office, in 2016, the area of shrimp farming was about 60% due to

climatic impacts as prolonged drought during the dry season, and more rainfall during wet season changed the water environment and temperature [32]. Long-term inactive fields dedicated for shrimp farming can create some environmental impacts, such as an increase in saline tolerance in soils, which can disperse and infect other land types, especially agricultural land in case of deficient irrigation systems, and inadequate land planning. Furthermore, because of proximity to the coastline, dry fields are considered to be sources of sand formation, based on changes in soil structure under severe conditions (such as prolonged high temperature, and oceanic winds). Such processes have dispersed sand from the coast inland.

Oceanic salt production started in Vietnam during the period of French colonization at the beginning of the 20th century. However, in Tuy Phong, it began in 1989, to take advantage of flat terrain, less rainfall, high levels of solar radiation, more than 300 sunny days per year, and local labor resources. Tuy Phong is one of four districts of Binh Thuan that has salterns. Salterns are in Vinh Hao commune, which covers 567.5 ha over 960 ha of total salt production in Binh Thuan [16]. Only a small portion of salterns was practiced in a small area of Phuoc The commune in 2001. Most lands currently used for salterns and shrimp fields were bare land and agricultural practicing in the past. Despite high yields from salt production, salt does not ensure higher incomes for salt farmers, generally a monthly income from 0.1 ha salt production is about $50 [16]. Figure 11 was taken at the area used for marine salt production in Phuoc The commune.

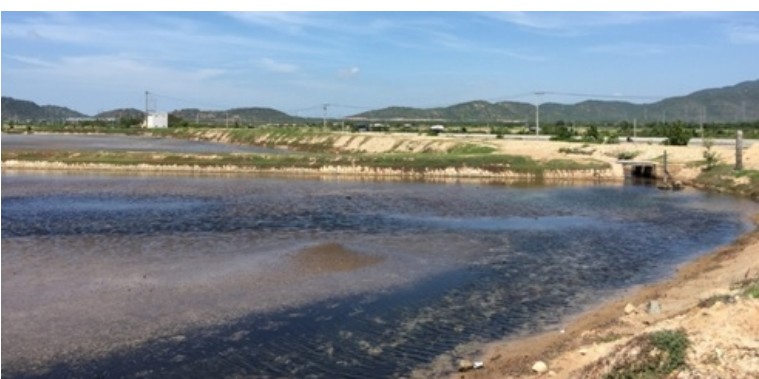

**Figure 11.** Salt production in Phuoc The commune.

Despite negative impacts of drought on sand spreading and land degradation, stimulation of salt production is a positive one of prolonged drought to the coastal area. Areas of salterns increased by nearly 60% after transitions of inactive agricultural lands to salt production. However, in similarity to shrimp farming, conversions to salterns may lead to significant environmental impacts. Without a commitment to sustainable practices, conversions to salt production can promote adverse effects such as saline intrusions into residential areas. This effect was found in Vinh Hao commune [11], where there were reports from local residents concerning influences of occurrence of salterns near their houses. Another drawback of salt production is economic value. Income from salt production is less than that from rice production, while its impacts on environment and human life are significant. Thus, there is more bias than gain.

### 4.3.3. Vegetative Areas

Impacts of human on land changes during drought events are both negative and positive. Permanently inactive agricultural land, deficit irrigation systems, unsustainable practices of salterns, and shrimp farming can affect local environments and soil. Despite negative effects, there are recognizable efforts of local residents and government, which were approached to support mitigation of drought, and expansion of sandy terrain. The most significant effort is vegetating bare soil and sand. Our findings show that vegetated area increased more than 70% (Table 4 and Figure 8), especially

ones located in urban and around agricultural land. In Phuoc The, this type of vegetation increased about 305 (bar graph in Figure 12). Plants can help to reduce urban heat islands and protect land from salinity process. Furthermore, vegetating sand can help to stabilize sand dunes near the coastline and prevent sand tracking. From our land survey in 2016, casuarina (Australian pine) is a common plant found along the study area's coastline, which can grow on salt-tolerant sand, and it can be constructed as a windbreak to prevent advances in sand.

### 4.4. Agricultural Practicing in Phuoc The Commune

Phuoc The commune experienced many changes in land use. The areal extent of Phuoc The commune is about 978 ha. In 2011, the agricultural area of Phuoc The was 288.87 ha covering 29.5% of total area, while salt and shrimp farming was 21% (approximately 206 ha). In 2016, there was a decrease in agricultural land, and a slight increase of salt and shrimp farming, 242.6 ha and 227.28 ha, respectively. During 2011–2016, vegetation cover increased from 60.8 ha to 90.59 ha, and a sandy plantation was 1ha larger in 2016 than in 2011 (17.29 and 18.15 ha), see table A2 in the Appendix B for more details. Figure 12 comprises LULC change map in Phuoc The with a bar graph reporting changes in some specific land types, and pie charts showing land proportions of agricultural, salterns and aquacultural lands

Drought led to a decline in agriculture and shrimp farming in this commune in 2016. In comparison to 2011, in 2016, areas of rice paddies reduced by 30% to change their active status to inactive because of reduced irrigation water. There is a significant increase in inactive shrimp farming due to drought, as higher surface temperatures have indirectly altered aquatic environments.

In Phuoc The, local residents transitioned paddies to orchards of dragon fruit or cassava in response to reductions in surface water. Although areas of orchards only rose 4% from 22% to 26% of total agricultural area, it can be shown that local people acknowledged drought impacts, and were trying to derive solutions for practicing agriculture under this adverse event, and to reduce economic losses. Furthermore, there was a transition from paddies to salterns. Most of those paddies were dry, distant from water resources, and close to industrial salt production areas. Transition to salterns is necessary to guarantee income for farmers in the case that no plants could be cultivated in neighborhood of salterns, and irrigation was not sufficient to prevent saline intrusions.

### 4.5. Limitations of the Study

While conducting our analysis and validation, we have observed several issues, such as the time gap between images, or disagreements between our results and the 2014 governmental map. Although there is three-week difference between two imagery datasets, the influence of acquisition time was not significant as a start of growing season (monsoon season). The 2014 map (shown in the Appendix C, Figure A1) is involved in our study to select training samples and validate our final results as a referenced map. Because we focus on drought and human impacts on some specific land types, such as agriculture (paddy fields, orchards or inactive agriculture). Thus, there is a significant difference of details on final maps compared to the reference map. This is an explanation for why, compared to 2014, built-up areas in 2011 and 2016 account for three fifths of 2014; 2014 agricultural area doubled, while its sand and water covered half in comparison to 2011 and 2016 (shown in Table 4).

Our findings are based on bi-temporal analysis using high-resolution data, a 2016 land survey, and Google Earth time series. They showed the changes in LULC in the study area of three coastal communes Binh Thanh, Lien Huong and Phuoc The, which are significant in agricultural lands and vegetated areas. Our findings attempted to reveal a prospectively undergoing story of LULC change in coastal areas in the North of Binh Thuan which were excluded in previous research [14,15,33]. We have discovered that there is a linkage of human and a dry condition (prolonged droughts) resulting in land transitions, especially in agricultural lands which were addressed in other similar studies [7,34]. However, because of bi-temporal analysis, the influence of drought is not significant; there could be a fluctuation on using land year by year. There is a study on drought impacts on LULC using time series

satellite data represented an argument that "no significant relationship between drought variations" and LULC change [35]. Nevertheless that study only carried out the meteorological drought (lack of rainfall) analysis, and excluded agricultural or hydrological drought. Meteorological drought may not affect agricultural lands if irrigation is proficient and sufficient [30] while agricultural and hydrological droughts have very close relationships to LULC upon available content of moisture, water resources and vegetation behavior during drought period. Further analysis will investigate identification of the main factors driving LULC changes in the study area; it is necessary to involve additional analysis and extension of time observation, such as monitoring available surface water during dry season, or a multiple temporal analysis for sequential images.

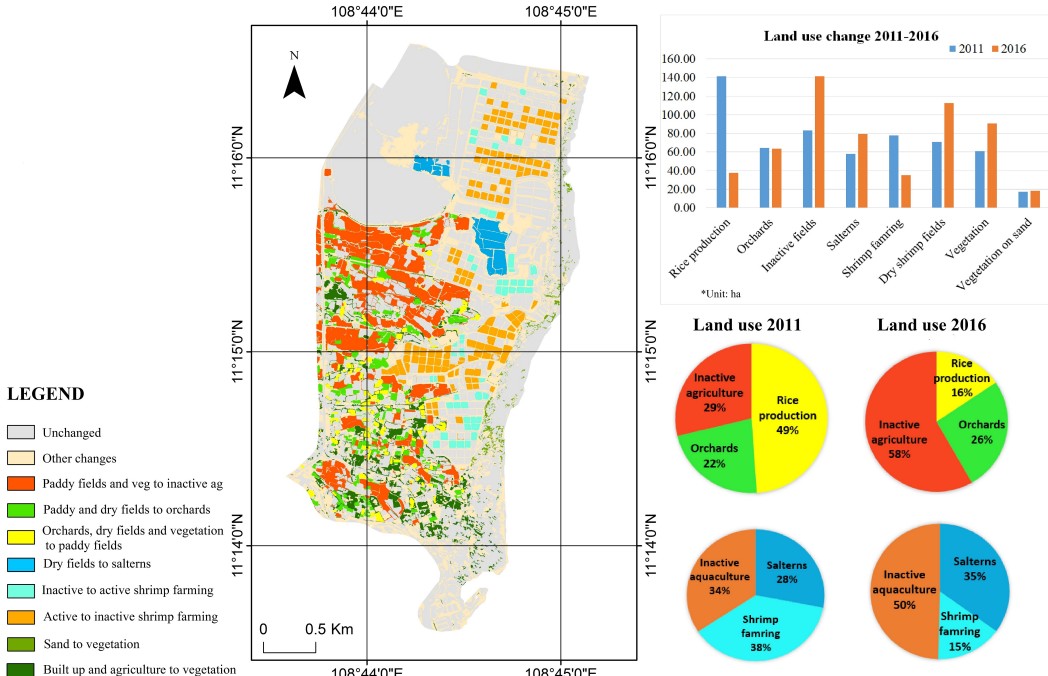

**Figure 12.** Land use change map of Phuoc The commune during 2011–2016. Bar graph and pie charts describe changes of eight specific classes regarding agriculture, shrimp farming and vegetation layers.

## 5. Conclusions

Prolonged drought can lead to many environmental and social impacts to affected areas. Tuy Phong, Binh Thuan is in a semi-arid area oriented parallel to a long coastline subject to severe droughts. This research is conducted to reveal impacts of drought on LULC changes in recent years (2011–2016), and to document some of these processes. In summary, recalling our research questions in the Introduction, our results have shown that:

(1) During drought periods, surface water resources reduced by one fourth, resulting in water shortage for irrigated and rain-fed rice fields, and a slight increase of bare land (unused land);

(2) To quantify impacts of drought, and water shortage on agricultural and aquacultural lands, we conducted a higher detailed image analysis; results showed a dramatic decrease of those land types (more than 50% and 44% , respectively). The extent of inactive agriculture and dry shrimp fields may promote land degradation, a spreading of sand into land, and desertification processes;

(3) Assessing interaction of local residents and droughts on LULC change, we observed local efforts to alternate rice and dry field by orchards growing drought-tolerant plants, such as dragon fruit or pitayas. Those plants not only adapt the prolonged dry conditions of the study area, but also bring higher income. Additionally, local residents and authorities were also active in promoting LULC transitions by vegetating bare soil and sand with stabilizing covering crops. Vegetating and crop rotation are examples of adaptive methods to combat drought.

Nevertheless, there are several limitations in our study, such as time gap between datasets that may result in vegetation variation in the beginning of growing season; a bi-temporal analysis cannot assess LULC variability regarding drought occurrence patterns; and there were discrepancies in our results compared to the map established by the local authorities in categorizing those fields as they were shown active status in the 2014 map. Therefore, a multi-temporal analysis will be the approach in our further research for improving our understandings of local land management in the study area under a scenario of prolonged drought. Accessing land management will include a procedure to quantify and qualify available water resources and effectiveness of irrigation systems, which secure fresh water to the entire study area during a drought period. During our summer field studies in 2016, we observed most agricultural lands near the coast shown as inactive in 2011 and 2016 maps were unable to support crop production. Reasons for the inactive status of those fields were long-term shortage of water, poor soil fertility, and soil salinity intrusion which resulted in land degradation, and promoted desertification in the study area. Thus, it is necessary to continue research to gain insight on human impacts on LULC, and on broader environmental processes during drought events. Such initiatives will support the development of solutions for land management and protection under threat of climatic hazards.

**Author Contributions:** H.T.T., J.B.C., R.H.W., and Y.S. conceived and designed experiments. H.T.T. performed the experiments, and analyzed data. S.V.P. contributed field research and reference materials. H.T.T., J.B.C., R.H.W. and Y.S. wrote the paper.

**Funding:** This research was sponsored by: (1) Imagery Grant by the Digital Globe Foundation, (2) Field study funded by Geography Department, Graduate School and the Interdisciplinary Graduate Education Program at Virginia Tech.

**Acknowledgments:** We would like to thank Digital Globe Courtesy for their imagery grant that provided valuable data to conduct this research. We acknowledge the Virginia Tech Graduate School, Virginia Tech's IGEP-RS (Interdisciplinary Graduate Education Program in Remote Sensing), and the Department of Geography for funding support for the summer 2016 field research. We acknowledge funding support by the Virginia Agricultural Experiment Station and the McIntire-Stennis Program of NIFA, United States Department of Agriculture. We are thankful to VT OASF (Virginia Tech Open Access Subvention Funding) support in our published article. We credit Virginia Tech's faculty—Steven Hodges, and National University of Singapore's faculty—Winston Chow as members of Hoa Tran's dissertation committee for their contributions to our research. We also recognize Devon Libby, the Digital Globe representation who supported us during our grant application process and his excellent communication during our project. We credit the Meteoblue team for providing free historical weather data of three locations: Lien Huong, Phan Thiet, and La Gi. We appreciate our field assistants: Ngoc Tran, and Ha Tran, who served as co-workers during the field research in Binh Thuan province. Lastly, we credit reviewers and the Remote Sensing journal's staff for their support during the processing procedure.

**Conflicts of Interest:** This research continues the previous study on drought severity monitoring and contributes to a Ph.D. dissertation. There are no conflicts of interest among authors.

## Appendix A. Training Sample

Table A1 describes characteristics of each class involved in the image classification procedure. Here we show samples of each land type from each sensor used in our study, displayed at consistent scales. Training samples were selected from the 2011 WorldView 2 (WV2), and the 2016 GeoEye 1 (GE1) images, at a scale of 1: 25,000. Some specific samples, such as buildings or roads, were at finer scale as noted under each image. Samples for the reference image were chosen from 2016 Google Earth scenes (Ref img).

**Table A1.** Training samples for WorldView 2 and GeoEye 1's classification procedures.

| Land Types | | | | Descriptions | | | Imagery Samples | | |
|---|---|---|---|---|---|---|---|---|---|
| *Level 1* | | *Level 2* | | *Shape & distribution* | *Color* | *Indices* | *WV2* | *GE1* | *Ref img* |
| 1 | Built up | 1 | Build up | Rectangle,dense, near main roads or coastlines. | White or light blue or red rooftops. | NDWI NHFD |  *1:5000 | | |
| 2 | Salterns and shrimp farming | 2 | Salterns certain size. | Rectangle, but white or brown. | Blue, dark blue NDWI | NDVI | | | |
| | | 3 | Active shrimp farming | Square, certain size.65 by 65 m. | Blue, dark blue. | NDVI | *1:10 000 | | |
| | | 4 | Inactive shrimp farming | Square, certain size 65 by 65 m. | Tan, brown, white. | NDSI | | | |
| | | 10 | Others land for aquaculture | Routes or rectangle. | Brown, tan. | NDSI NHFD | *1:5000 | | |
| 3 | Water | 5 | Water | Undefined shape. | Dark blue or green. | NDVI | | | |

**Table A1.** *Cont*.

| Land Types | | | Descriptions | | | Imagery Samples | | |
|---|---|---|---|---|---|---|---|---|
| *Level 1* | *Level 2* | | *Shape & distribution* | *Color* | *Indices* | *WV2* | *GE1* | *Ref img* |
| 4 Agriculture | 6 | Rice production | Undefined shape, smooth surface. | Dark to light green (natural color).<br><br>Dark to bright red (false color). | NDVI | | | |
| | 11 | Orchards | Rectangle, rows& and columns,mixed with built up. | Green and/or mixing brown as soil background. | NDVI | | | |
| | 12 | Inactive agriculture | Rectangle or undefined shape. | Brown, tan with some greenness | NDSI | | | |
| 5 Vegetation | 7 | Vegetation | Undefined shape, narrow like borders. | Green to dark green | NDVI | | | |
| 6 Bare land | 8 | Bare land | Undefined shape, no or very spare vegetation. | Brown, tan, some greenness. | NDSI | | | |
| 7 Sand | 9 | Sand | Undefined shape, near coastlines, or dried fields. | Yellow or white, tan, some greyness. | NDVI | | | |
| | 13 | Vegetation on sand | Undefined shape, near coastlines. | Green | NDVI | | | |

## Appendix B. Land Use Change in Phuoc The community

Statistical data for analyzing LULC change in Phuoc The community—Table A2.

**Table A2.** Land use transition of Phuoc The from 2011 to 2016 (unit: ha) of 13 land use types. Gray represents unchanged land types; other colors refer to significant changes of some specific land types shown in Figure 12.

| 2016<br>2011 | 1 | 2 | 3 | 4 | 5 | 6 | 7 | 8 | 9 | 10 | 11 | 12 | 13 |
|---|---|---|---|---|---|---|---|---|---|---|---|---|---|
| 1 | 46.95 | 0.65 | 0.01 | 0.03 | 0.49 | 0.48 | 11.92 | 7.09 | 0.9 | 1.35 | 1.90 | 2.04 | 0.69 |
| 2 | 0.72 | 47.91 | 1.13 | 1.71 | 0.41 | 0.02 | 0.46 | 1.178 | 0.1 | 2.62 | 0.9 | 0.04 | 0.11 |
| 3 | 0.26 | 0.9 | 10.01 | 53.89 | 0.06 | 0 | 0.96 | 0.54 | 0.04 | 9.97 | 0 | 0.11 | 0.15 |
| 4 | 0.22 | 0 | 18.28 | 39 | 0.02 | 0 | 0.79 | 0.25 | 0.15 | 12.01 | 0 | 0.01 | 0.05 |
| 5 | 0.35 | 0.09 | 0.02 | 0.04 | 6.9 | 0.22 | 2.41 | 1.8 | 0.13 | 0.38 | 0.06 | 0.23 | 0.12 |
| 6 | 2.39 | 0.21 | İ2 | 0.05 | 0.008 | 18.8 | 11.44 | 2.55 | 0 | 0.12 | 15.77 | 89.69 | 0 |
| 7 | 6.68 | 0.89 | 0.27 | 0.73 | 0.52 | 3.46 | 25.16 | 6.15 | 0.02 | 2.02 | 5.74 | 9.1 | 0.06 |
| 8 | 16.12 | 1.3 | 0.11 | 0.04 | 1.97 | 0.32 | 5.46 | 140.0 | 2.62 | 0.31 | 0.72 | 1.43 | 0.44 |
| 9 | 6.88 | 0.36 | 0.26 | 0.12 | 0.41 | 0 | 1.22 | 1.29 | 47.5 | 0.85 | 0.02 | 0.04 | 5.94 |
| 10 | 3.86 | 11.06 | 4.047 | 16.38 | 0.42 | 0 | 2.27 | 3.16 | 0.52 | 38.79 | 0.01 | 0.02 | 0.89 |
| 11 | 2.65 | 0 | 0 | 0.02 | 0 | 8.08 | 20.04 | 2.12 | 0 | 0.03 | 26.51 | 5.12 | 0 |
| 12 | 1.83 | 15.05 | 0.66 | 0.76 | 0.02 | 6.42 | 7.79 | 4.18 | 0 | 1.02 | 11.89 | 33.46 | 0 |
| 13 | 0.52 | 0.05 | 0.09 | 0.04 | 0.11 | 0 | 0.67 | 0.15 | 5.77 | 0.17 | 0 | 0.02 | 9.7 |

## Appendix C.  The 2014 Land Use Land Cover Map

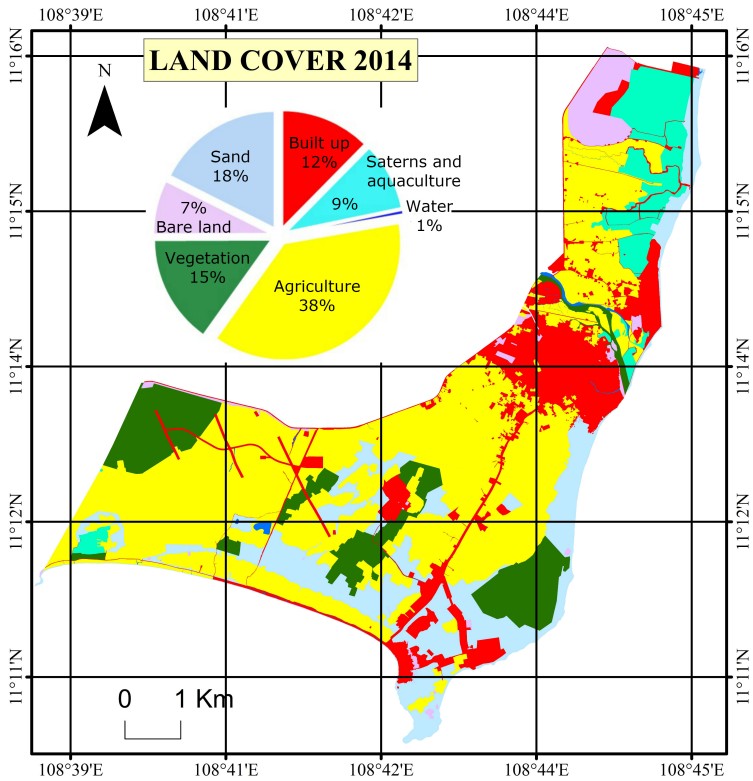

**Figure A1.** The governmental map of land use in the study area in 2014.

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
