# Peer review of "Drought and Human Impacts on Land Use and Land Cover Change in a Vietnamese Coastal Area"

_remotesensing, doi:10.3390/rs11030333_

Round 1

Reviewer 1 Report

The study analyzed the land use/land cover (LULC) changes in three communes of the Vietnamese coastal area due to the effect of frequent drought events.  Authors applied an object-based classification of the high-resolution imageries: WorlView2 for 2011 and the GeoEye1 for 2016. Their results showed relevant changes in the landcover between the two dates, and these changes were attributed to drought events.

Major issues.

As you are analyzing the LULC change due to drought events, these events need to be very well defined, both spatial and temporal. In Fig. 4 you showed the maps for three drought events, 2013-2014, 2014-2015, and 2015-2016. However, you didn't indicate how these events were defined, i.e., by rainfall deficit (and for how long), by hydrological deficit (and for how long), or vegetation impact. Also, the map doesn't fit with the one showed in Fig. 3, and the communes are not readily identifiable. To support your results and conclusions, the reader must know what kind of drought are you talking about and based on which parameter (e.g., rainfall, soil moisture) and which aggregated time did you used. To achieve this, there is a wide range of drought indices that could be used such as the Standardized Precipitation Index (SPI; McKee 1993), the Standardized Precipitation Evapotranspiration Index (SPEI; Vicente-Serrano et al., 2010) Vegetation Condition Index (VCI, Kogan 1995). Having well defined the drought event (spatially and temporally) you could compare specific drought intensities with LULC, and then you could attribute changes in LULC to drought events. Probably, some of the variability on LULC  may be explained due to different levels of drought intensity. For example, the variation of LULC changes from 2011 to 2016 for rice production might be associated in a change of X% of the intensity in the drought index Y (i.e., SPI3), and so on for the remaining land cover types.

McKee, T. B., Doesken, N. J., & Kleist, J. (1993). The relationship of drought frecuency and duration to time scales. In In Proceedings of the International 8th Conference on Applied Climatology. American Meteorological Society, Anaheim, CA, USA, 17-22 January (pp. 179–184).

Vicente-Serrano, S. M., Beguería, S., & López-Moreno, J. I. (2010). A Multiscalar Drought Index Sensitive to Global Warming: The Standardized Precipitation Evapotranspiration Index. Journal of Climate, 23(7), 1696–1718. https://doi.org/10.1175/2009JCLI2909.1

Kogan, F. N. (1995). Droughts of the Late 1980s in the United States as Derived from NOAA Polar-Orbiting Satellite Data. Bull. Am. Metor. Soc., 76(5), 655–668. https://doi.org/10.1175/1520-0477(1995)076<0655:DOTLIT>2.0.CO;2

Regard of the structure of the manuscript, it's not adequately defined. The introduction has paragraphs that are more suited to the study area section (L52-73). The text written between L52-63  correspond to the methodology. Furthermore, most of the figures are not defined, even not mentioned in the text. The manuscript lacks a discussion section in which the results could be analyzed in perspective and related to other similar studies.

Minor issues

Figures 5, 9, and 11 not suited for a research article. Which is their purpose? How do these photos help to understand the methodology or results better?

Maps on Figure 8 (top) should be bigger to be more readable, and a table could replace the pie charts at the bottom. Also, lacks letters identifying each map or plot.

Figures captions are poor described, must be improved. You need to include letters (a,b,c,d) in Figure 10 (and in any other having multiple plots or maps). The lines graph in Figure 10 is not adequade, due that these aren't continuous variables.

Author Response

Dear reviewer,

Your comments and suggestions are valuable to us.

We have made edits and changes regarding your suggestions to illustrate our research ideas and report relevant results and discussion. 

Please find our responses to your comments in the attachment.

We are looking forward to hearing your feedback soon.

Many thanks

Reviewer 2 Report

The manuscript titled "Drought and Human Impacts on Land Use and Land Cover Change in a Vietnamese Coastal Area" it seems improved after resubmssion. After reading the manuscript the questions and comments are as follow:

- Please make a legend or explain in text about Fig.2.

- Please remove the background color of flowchart Fig.6 and 7

- How you make decisions for 13 classes number? and why has 2 types? is the class definition has special sources?

- For image classification using the segmentation and decision tree, how you decide about threshold value or just used the indexes?

- How about image classification result validation? did you used Google Earth image? 

- The conclusion is not well supported the result and research goal, please make clear the result and finding of research

Author Response

Dear reviewer,

We are thankful for your comments on our manuscripts.

Please find our responses in the attached file. 

We are looking forward to hearing you back soon.

Respectfully

Authors

Reviewer 3 Report

In Abstract, drought definition at the very first sentence needs to rewrite. Drought is not always due to the sudden shortage of rainfall. Coastal regions of where? The abstract must rewrite and include the country name too.

Need to add few references to the introduction section, apart from the two citations used. Figure 01, must show Vietnam and mark the study area on it.

Justify your statement (line 78), 60% of the area subject to desertification? Figure 02, use a different colour for Agriculture, for people who wish to print your paper in BW.

For line 140, use the average precipitation figures of Vietnam to explain dryness of your study area. Figure 3 maps are not clear at all. Re does the figure to be clear at 100% display. Also, explain the drought classification used in figure 4.

Classification and validation process is acceptable. However, justify the selection of 61 ground references points. Why 61? Also, the very high growth of vegetation category even in drought-affected time, need to explain further. You said, you “believe”, but such a statement is not strong enough. Also, the LCLU changes within the considered time duration may be a sample of longer fluctuations of the pattern. Discuss that possibility too.

Pay attention to the figures. Some are too small to view at 100% display.

Author Response

We appreciate your insightful comments on your manuscript.

Please find our responses in the attachment. 

We are looking forward to hearing your feedback soon.

Many thanks.

Reviewer 4 Report

Using high resolution satellite data, this study examined the impacts of drought and human actions on the land use and land cover change in the coastal areas of Vietnam.  While the study is interesting, I believe it should be revised significantly based on my comments below in order for it to be published in the journal Remote Sensing:

1)      The introduction is quite disorganized and at times seems repetitive. Please re-write so that it flows smoothly as if telling a story.

2)      Incorporate Sections 1.1 and 1.2 into the introduction section and make it short and concise.

3)      Figure 1 should be moved to the study area section.

4)      Please re-write section 3.3 in a paragraph format rather than in bullet format.

5)      Please explain some of the criteria used for the decision tree in Figure 7.  For example, why were areas more than 1000 (in whatever units) used for the separation of water and built-up area? 

6)      In line 250, I am a bit confused about the 460 random points and the 61 ground reference points. Can you please explain how 61 ground reference points were used to examine the accuracy of the 460 random points?  Is it 61 ground reference points per land use/cover class?

7)       Section 4.2 is a bit confusing and difficult to understand.  Please revise this section.

8)      The legends of the map in figure 8 are not readable. Please produce an image with higher resolution.

9)      Please discuss how the results of this study compare to other previous studies in other parts of the world.

10)   Please discuss the limitations of the study (if any).

Author Response

We appreciate your insightful comments on our manuscript.

We have made edits to improve our texts.

Please find our responses in the attachment.

We are looking forward to hearing your feedback soon.

Many thanks

Round 2

Reviewer 1 Report

Thanks for the revised version of the manuscript. The authors addressed all my comment, and I think the document improved and is more precise for readers.

Some minor issues.

Figure 1 it is not mentinoed in the text. Would be more adequate to put this figure in the section study area.

L55-70: This paragraph should be shortener, too much information about the study area and the methodology that is redundant. Would be better if this paragprah was briefly focus in the general and specifi goal.

L71-72: This looks better inline rather than as a list.

L74-87: This is methodology.

L93: this is not the proper reference. Please look at Kottek et al. (2006)

Kottek, M., J. Grieser, C. Beck, B. Rudolf, and F. Rubel, 2006: World Map of the Köppen-Geiger climate classification updated. Meteorol. Z., 15, 259-263. DOI: 10.1127/0941-2948/2006/0130.

In section 3.2, the first paragraph is no understandable. Did you used 2014 for training and 2016 for validation?

Author Response

Dear reviewer,

We are thankful for your insightful comments and suggestions.

We have improved our manuscript.

Please find our responses in the attachment. 

We are looking forward to hearing you back soon.

Respectfully

Authors

Reviewer 4 Report

I am satisfied with the revisions made by the authors and I recommend it for publication. 

Author Response

Dear reviewer,

We appreciate your comment.

We wish you all bests.

Respectfully

Authors